# (GIGA)bYte

DATA RELEASE

# Utilizing a chromosomal-length genome assembly to annotate the Wnt signaling pathway in the Asian citrus psyllid, *Diaphorina citri*

Chad Vosburg[1,2], Max Reynolds[1], Rita Noel[1], Teresa Shippy[3], Prashant S. Hosmani[4], Mirella Flores-Gonzalez[4], Lukas A. Mueller[4], Wayne B. Hunter[5], Susan J. Brown[3], Tom D'Elia[1] and Surya Saha[4,6,*]

1 Indian River State College, Fort Pierce, FL 34981, USA
2 Department of Plant Pathology and Environmental Microbiology, Pennsylvania State University, University Park, PA 16802, USA
3 KSU Bioinformatics Center, Division of Biology, Kansas State University, Manhattan, KS 66506, USA
4 Boyce Thompson Institute, Ithaca, NY 14853, USA
5 USDA-ARS, US Horticultural Research Laboratory, Fort Pierce, FL 34945, USA
6 Animal and Comparative Biomedical Sciences, University of Arizona, Tucson, AZ 85721, USA

## ABSTRACT

The Asian citrus psyllid, *Diaphorina citri*, is an insect vector that transmits *Candidatus* Liberibacter asiaticus, the causal agent of the Huanglongbing (HLB), or citrus greening disease. This disease has devastated Florida's citrus industry, and threatens California's industry as well as other citrus producing regions around the world. To find novel solutions to the disease, a better understanding of the vector is needed. The *D. citri* genome has been used to identify and characterize genes involved in Wnt signaling pathways. Wnt signaling is utilized for many important biological processes in metazoans, such as patterning and tissue generation. Curation based on RNA sequencing data and sequence homology confirms 24 Wnt signaling genes within the *D. citri* genome, including homologs for beta-catenin, Frizzled receptors, and seven Wnt-ligands. Through phylogenetic analysis, we classify *D. citri* Wnt ligands as *Wg/Wnt1*, *Wnt5*, *Wnt6*, *Wnt7*, *Wnt10*, *Wnt11*, and *WntA*. The *D. citri* version 3.0 genome with chromosomal length scaffolds reveals a conserved *Wnt1-Wnt6-Wnt10* gene cluster with a gene configuration like that in *Drosophila melanogaster*. These findings provide greater insight into the evolutionary history of *D. citri* and Wnt signaling in this important hemipteran vector. Manual annotation was essential for identifying high quality gene models. These gene models can be used to develop molecular systems, such as CRISPR and RNAi, which target and control psyllid populations to manage the spread of HLB. Manual annotation of Wnt signaling pathways was done as part of a collaborative community annotation project.

**Submitted:** 18 December 2020

* Corresponding author. E-mail: suryasaha@cornell.edu

Preprint submitted at https://doi.org/10.1101/2020.09.21.306100

**Subjects** Genetics and Genomics, Animal Genetics, Bioinformatics

## DATA DESCRIPTION

### Introduction

*Diaphorina citri* (NCBI:txid121845) is the insect vector of Huanglongbing (HLB), or citrus greening disease, which has devastated global citrus production [1, 2]. HLB management is heavily based on controlling the spread of *D. citri*. To better understand the insect's

biology, the *D. citri* genome has been manually annotated to curate accurate gene model predictions. Accurate gene models can be used to develop novel insect control systems that utilize molecular therapeutics such as CRISPR (clustered regularly interspaced short palindromic repeats) and RNA interference (RNAi) to control the spread of *D. citri* [3, 4]. These molecular therapeutics would be gene-specific, thus would reduce reliance on broad-spectrum insecticides that have given rise to resistant *D. citri* populations [5–7].

## Context

Here, we report *D. citri* genes involved in both canonical and noncanonical Wnt signaling. Wnt signaling is important for many biological processes in metazoans, such as patterning, cell polarity, tissue generation, and stem cell maintenance [8–10]. In the model insects *Drosophila melanogaster* and *Tribolium castaneum*, knockout and knockdown of Wnt ligands and other Wnt signaling components have detrimental effects on embryo development and adult homeostasis [11–16]. Wnt signaling components could therefore be effective knockout targets to limit the spread of *D. citri*, thus reducing HLB incidence. We curated a comprehensive repertoire of Wnt signaling genes in *D. citri*. Twenty-four gene models corresponding to canonical and noncanonical Wnt signaling genes have been annotated, including seven Wnt ligands, three *frizzled* homologs, *arrow*, *armadillo/beta-catenin*, and receptor tyrosine kinases *ROR* and *doughnut*. We were unable to find *Wnt8/D*, *Wnt9*, and *Wnt16* as well as *Wnt2-4*, which have been lost in insects. The mechanisms of Wnt signaling appear to be mostly conserved and comparable to those found in *D. melanogaster* (Table 1). A model for canonical Wnt signaling in *D. citri* based on curated genes is shown (Figure 1). This is an important first step towards understanding critical biological processes that might be targeted to control the spread of *D. citri*, and may provide broader insights into the mechanisms of Wnt signaling in this important hemipteran vector.

## METHODS

We used the psyllid genome curation workflow used for community annotation (Figure 2) [17].

To summarize, orthologous protein sequences for Wnt pathway genes were collected from the National Center for Biotechnology Information (NCBI) (RRID:SCR_006472) protein database [18] and used to BLAST (RRID:SCR_004870) search the *D. citri* MCOT transcriptome database [19]. The MCOT transcriptome is a transcriptome assembly utilizing Maker (RRID:SCR_005309), Cufflinks (RRID:SCR_014597), Oases (RRID:SCR_011896), and Trinity (RRID:SCR_013048) pipelines to provide a comprehensive set of predicted gene models. High-scoring MCOT models (accessions available in Table 2) were then searched on the NCBI protein database using NCBI BLAST to confirm the viability of the predicted MCOT models. The high-scoring MCOT models that had promising NCBI search results were used to search the *D. citri* genome. Genome regions containing computationally predicted gene models with high sequence identity to the query sequence from the MCOT transcriptome were investigated within JBrowse (RRID:SCR_001004). Gene models were modified using the Apollo (RRID:SCR_001936) gene annotation platform, based on mapped DNA-Seq, RNA-Seq, Iso-Seq, orthologous proteins, and other lines of evidence to edit and confirm manual annotations and gene structure. The gene models were analyzed with NCBI BLAST to assess

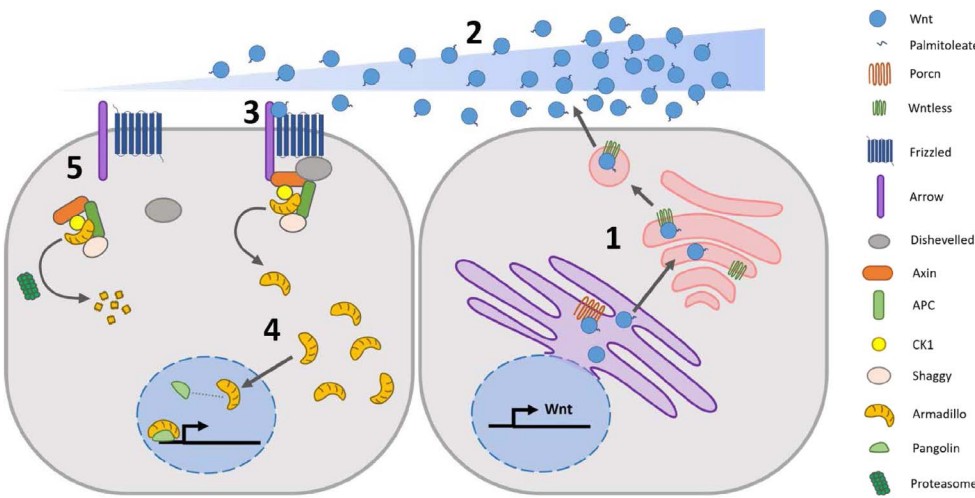

**Figure 1.** Theoretical model of canonical Wnt signaling cascade in *D. citri* based on curated genes. (1) Wnt is secreted. (2) Wnt concentration gradient forms. (3) Wnt binds to Frizzled and releases Armadillo. (4) Armadillo migrates into the nucleus, associates with transcription factor Pangolin, and regulates gene expression. (5) Armadillo is degraded in the absence of Wnt.

**Table 1.** Summary of gene copy numbers in various model insect species, including *Diaphorina citri*. Wnt pathway ortholog numbers in five different insect species. *Drosophila melanogaster, Apis mellifera, Tribolium castaneum*, and *Acyrthosiphon pisum* copy numbers were determined using Flybase, OrthoDB, NCBI Genbank, Uniprot, and several other publications [15, 20–22]. *Diaphorina citri* numbers represent the number of manually annotated genes in the *D. citri* v3.0 genome.

| Gene | *Drosophila melanogaster* | *Apis mellifera* | *Tribolium castaneum* | *Acyrthosiphon pisum* | *Diaphorina citri* v3 |
|---|---|---|---|---|---|
| *Wnt1* | 1 | 1 | 1 | 1 | 1 |
| *Wnt5* | 1 | 1 | 1 | 1 | 1 |
| *Wnt6* | 1 | 1 | 1 | 0 | 1 |
| *Wnt7* | 1 | 1 | 1 | 1 | 1 |
| *Wnt8/D* | 1 | 0 | 1 | 0 | 0 |
| *Wnt9* | 1 | 0 | 1 | 0 | 0 |
| *Wnt10* | 1 | 1 | 1 | 0 | 1 |
| *Wnt11* | 0 | 1 | 1 | 1 | 1 |
| *Wnt16* | 0 | 0 | 0 | 1 | 0 |
| *WntA* | 0 | 1 | 1 | 1 | 1 |
| *Pangolin* | 1 | 1 | 1 | 1 | 1 |
| *Armadillo* | 1 | 1 | 2 | 2 | 1 |
| *Wntless* | 1 | 1 | 1 | 1 | 1 |
| *Porcupine* | 1 | 1 | 1 | 1 | 1 |
| *Derailed* | 2 | 1 | 0 | 1 | 1 |
| *Doughnut* | 1 | 1 | 1 | 1 | 1 |
| *Arrow* | 1 | 1 | 1 | 1 | 1 |
| *Frizzled* | 4 | 2 | 3 | 2 | 3 |
| *ROR* | 2 | 2 | 3 | 2 | 2 |
| *Dishevelled* | 1 | 1 | 1 | 1 | 1 |
| *Shaggy* | 1 | 1 | 1 | 2 | 1 |
| *Axin* | 1 | 1 | 1 | 1 | 1 |
| *ck1-gamma* | 1 | 1 | 1 | 1 | 1 |
| *Apc* | 2 | 1 | 1 | 1 | 1 |

their completeness. MUSCLE (RRID:SCR_011812) multiple sequence alignments of the *D. citri* gene model sequences and orthologous sequences were created through MEGA7

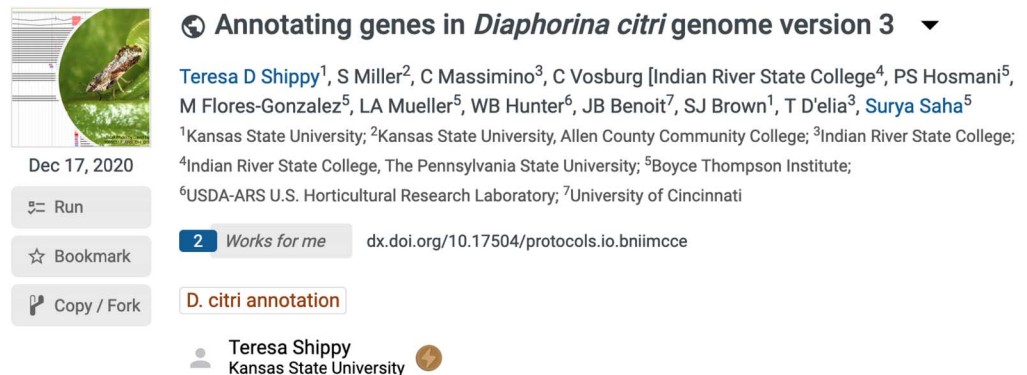

**Figure 2.** Protocol for psyllid genome curation [17]. https://www.protocols.io/widgets/doi?uri=dx.doi.org/10.17504/protocols.io.bniimcce

(RRID:SCR_000667) [23]. Neighbor-joining trees were constructed using MEGA7 with p-distance for determining branch length and 1000 bootstrapping replications to measure the precision of branch placement. In special cases, phylogenetic analysis in conjunction with NCBI BLAST scores was used to properly name and characterize the manually annotated gene models.

RNA-seq data from whole body adults and nymphs raised on *C. medica* and *C. sinensis* are available from NCBI BioProject PRJNA609978. We used proteins from *Drosophila melanogaster* (*Dm*) [24], *Tribolium castaneum* [25], *Bombyx mori* [26], *Apis mellifera* [27], *Nasonia vitripennis* [28], *Acyrthosiphon pisum* [29], *Nilaparvata lugens* [30, 31], *Sipha flava* [32], *Halyomorpha halys* [33], *Cimex lectularius* [34], *Aedes aegypti* [35], *Anopheles gambiae* [36], *Branchiostoma floridae* [37], *Penaeus vannamei* [38], *Folsomia candida* [39], *Spodoptera litura* [40], *Homo sapiens* [32] and *Oncopeltus fasciatus* [34, 41].

## Data validation and quality control

The loss of Wnt ligand genes is more common in insects than in other metazoans [20], which leads to a highly variable array of *Wnt* genes and Wnt signaling components from species to species [15, 21, 22, 42]. We performed a phylogenetic analysis to characterize the *D. citri* Wnt repertoire (Figure 3). The ortholog sequences used for this analysis were collected from the NCBI protein database [18]; see the 'Availability of Data and Materials' section for accession numbers. Seven *D. citri Wnts* were identified and classified as *Wnt1* (also known as *wingless*), *Wnt5*, *Wnt6*, *Wnt7*, *Wnt10*, *Wnt11*, and *WntA* (Figures 3 and 4). In comparison, seven *Wnt* genes have been identified in *D. melanogaster*, nine in *T. castaneum*, and six in *Acyrthosiphon pisum* [22, 42]. The collection of *Wnt* genes found in *D. citri* is like that found in other insects, and no *Wnt* subfamilies have been identified as being unique to *D. citri*. Contrary to previous reports [43], *D. citri* does appear to possess a *Wnt6* gene.

*Wnt1*, *Wnt6*, and *Wnt10* typically occur close together in a highly conserved gene cluster [44, 45]. The chromosomal length genome assembly in v3.0 suggests that this cluster is also conserved in *D. citri*, located at a position between 26.4 Mb (megabases) and 26.6 Mb on scaffold 4 (i.e. chromosome 4) [46]. In comparison, the *Wnt1-6-10* cluster is located at a position between 7.30 Mb and 7.38 Mb on chromosome 2L of *D. melanogaster*, and between 5.50 Mb and 5.53 Mb on linkage group 5 in *T. castaneum*. The only gene from this cluster

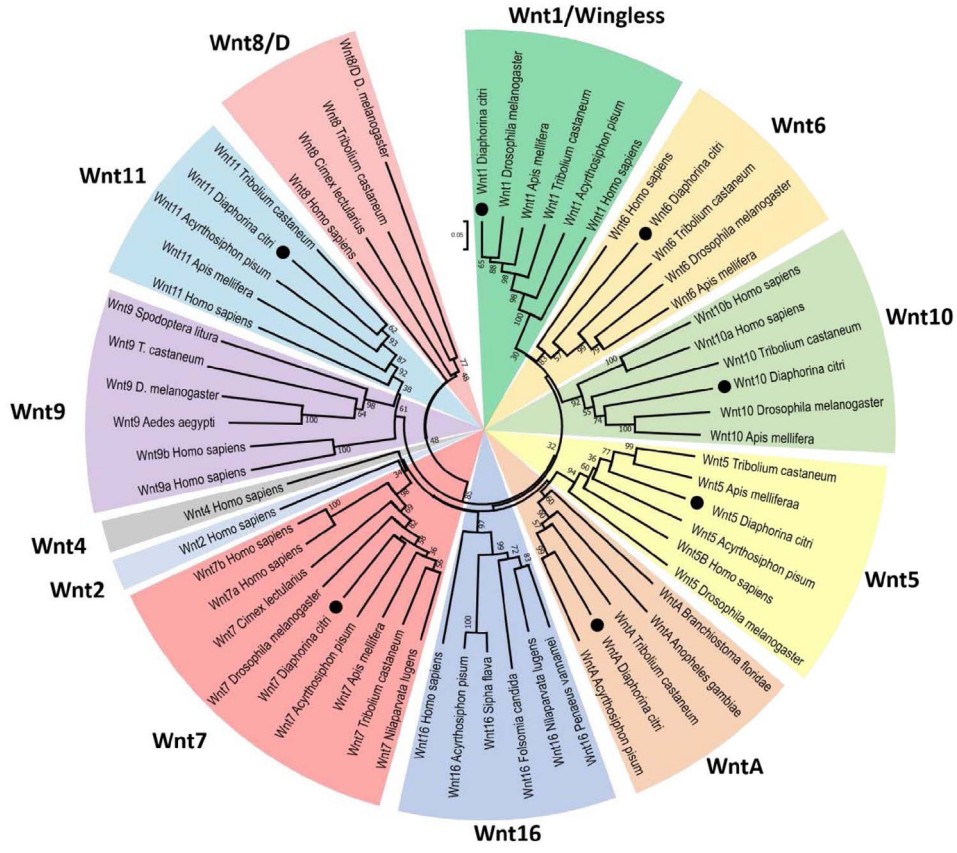

**Figure 3.** Neighbor-joining tree of Wnt protein sequences. Phylogenetic analysis was performed to categorize the seven *D. citri Wnt* genes (signified by dots). Wnt families are distinguished by clades and are color coded. Bootstrap values are based on 1000 replicates and values under 25 are removed. Ortholog sequences were collected from NCBI protein database [18]. Analysis was performed using MEGA7 [23].

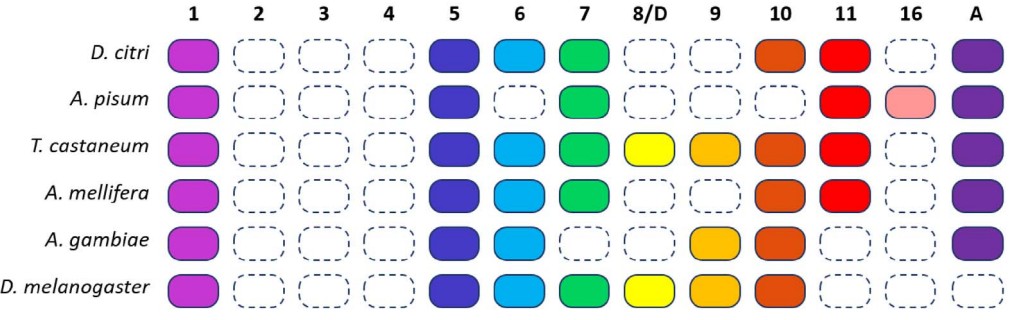

**Figure 4.** *Wnt* genes in six insects. A colored box indicates the presence of a *Wnt* subfamily (1–11, 16, and A) in that insect, while a white box indicates the loss of a subfamily. For example, all six species have *Wnt1* and *Wnt5*, none have *Wnt2-4*, and only *A. pisum* has *Wnt16*. Homologs of *Wnt8* in *T. castaneum* and *D. melanogaster* are also referred to as *WntD*.

present in *A. pisum* is *Wnt1*, which is located on the X chromosome. The close phylogenetic relationship of *Wnt1*, *Wnt6*, and *Wnt10* in *D. citri* (Figure 3) supports the hypothesis that this cluster is the result of an ancient duplication event, one that may predate the



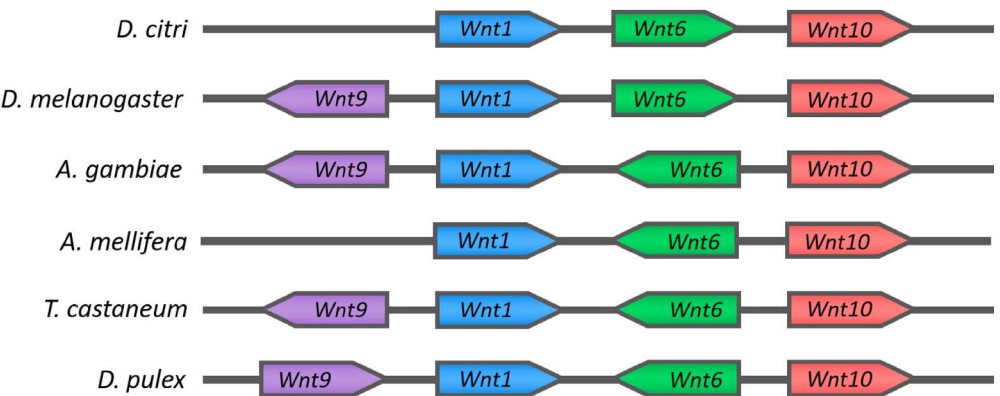

**Figure 5.** *Wnt1-6-10* cluster comparison. Organization of the *Wnt1-6-10* cluster in *D. citri* is similar to that of *D. melanogaster* and differs from what may be a basal arthropod gene arrangement seen in *A. gambiae, T. castaneum, A. mellifera,* and *D. pulex*. Gene lengths are not to scale.

divergence of cnidarians and bilaterians [45]. The orientation of these clustered *D. citri Wnt* genes is like that found in *D. melanogaster* and differs from what may be a basal arthropodal organization of *Wnt*s found in species of Coleoptera, Hymenoptera, and Cladocera (Figure 5). *Wnt9* is also associated with this gene cluster when present in the genome. However, as with *A. pisum, Wnt9* was not found in the *D. citri* genome and appears to have been lost during evolution. A second *Wnt* cluster, *Wnt5* and *Wnt7*, is also common among non-insect metazoans. This cluster is not seen in *D. citri*; however, *D. citri Wnt5* and *7* are located relatively close to one another (within 220 Kb [kilobase pairs]) on scaffold 13 (i.e. chromosome 13).

The mechanisms that act to conserve these *Wnt* gene clusters are not well understood. In the basal metazoan *Nematostella vectensis*, clustered *Wnt* genes do not exhibit similar expression patterns or *Hox*-like collinearity [44], and may not share regulatory elements. Whole body transcript expression data from egg, nymph, and adult stages [47] obtained from the Citrus Greening Expression Network (CGEN) [48] shows varying levels of expression among the clustered genes in different life stages of *D. citri* (Figure 6). However, it appears that *Wnt1* and *10* are similarly upregulated during embryonic psyllid development and downregulated during the adult stage. Similar transcript levels of *Wnt1* and *6* are seen in the nymphal stage. This suggests there may be shared regulation dependent upon life stage. Furthermore, ordering within the clusters is subject to rearrangement (Figure 5) [42, 44]. This may indicate that gene directionality is not a factor in conserving this cluster. Our annotation findings support the hypothesis that the *Wnt1-6-10* cluster is being preserved through either natural selection or an unknown mechanism. A better understanding of the regulatory hierarchy controlling *Wnt* expression might shed light on the significance of *Wnt* gene associations in the genome. Future characterization of the coding and noncoding regions surrounding these *D. citri Wnt*s (e.g., tandem repeat analysis) could also provide more insight into the mechanisms causing *Wnt* duplication events.

The organization of the genomic reference sequence into chromosomal length scaffolds was essential for revealing *D. citri* gene clustering. Because of their shorter scaffold lengths, previous genome assemblies were often unsupportive in confirming the proximity of genes.

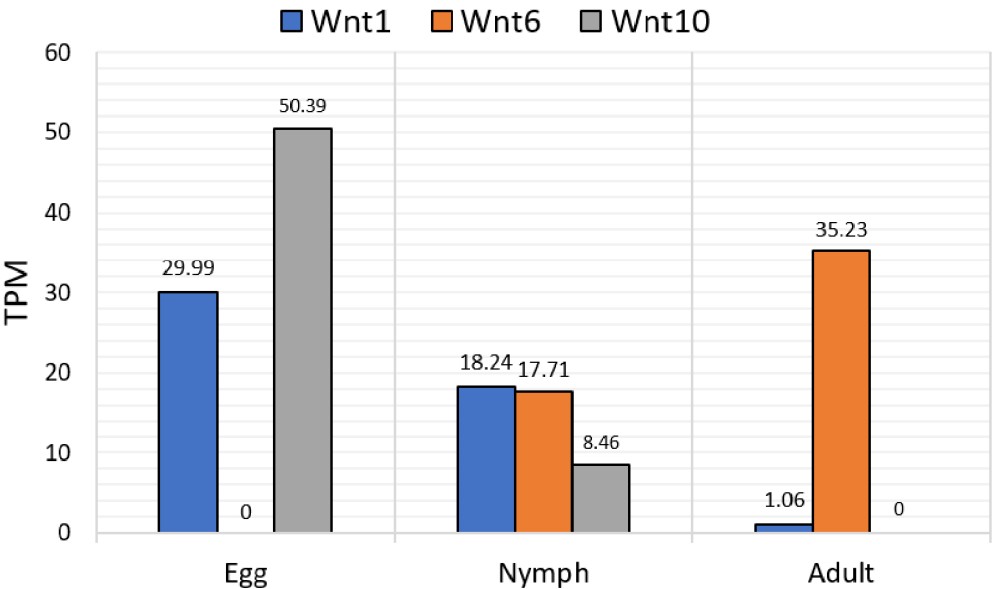

**Figure 6.** Transcript levels of clustered *Wnt* transcripts during different *D. citri* life stages. Whole body transcript expression data from egg, nymph, and adult stages [47] were collected from CGEN [48]. The psyllids were raised on *Citrus macrophylla* and were not infected with *Candidatus* Liberibacter asiaticus. Expression values shown in transcripts per million (TPM).

Genome v2.0 assembly errors had likely misrepresented the location of *Wnt10*, making it appear to be separated from *Wnt1* and *Wnt6*. A complete *Wnt1-6-10* cluster was found in the improved chromosome length assembly v3.0. Thus, the quality of the reference genome should be considered when performing phylogenetic studies.

Orthologs for *Wnt2*, *Wnt3*, *Wnt4*, *Wnt8/D*, *Wnt9*, and *Wnt16* were not located in the *D. citri* genome. The close identity of certain *Wnt* subfamilies makes it difficult to distinguish between them; however, the loss of *Wnt2–4* is expected because they are absent in all insects [20]. *Apis mellifera* and the hemipteran *A. pisum* have been reported to lack *Wnt8/D*. Perhaps this *Wnt* subfamily has been lost in the divergence from other insect groups [22]. Additionally, *Wnt16* was not found in *D. citri* v3.0. This finding contrasts with the gene predictions of other hemipteran genomes available at NCBI, namely *A. pisum, Sipha flava*, and *Nilaparvata lugens* (Figure 3). Based on whole body RNA expression data collected from CGEN, *Wnt6* has the highest average transcript levels of all the *Wnt* genes in both nymph and adult psyllids (Figure 7). The relatively high number of transcripts suggests that *Wnt6* is important during both metamorphosis and adult stage homeostasis, and may be a good knockout target for molecular therapeutics. Transcript expression of *Wnt6* in adults is mainly concentrated in the legs and thorax, averaging 102 transcripts per million (TPM) and 272 TPM, respectively. This is considerably higher than all other *Wnt* genes in these tissues, which only average between 0.26 and 3.00 TPM. It is unclear if other *Wnts* can be upregulated to compensate for the loss of *Wnt6*. Perhaps targeting multiple *Wnt* genes, or the mechanisms by which Wnt is secreted (i.e. Porcupine and Wntless), would be more disruptive to *D. citri* physiology.

Several receptors and co-receptors associated with canonical and noncanonical signaling have been identified (Table 2). Three paralogs for the Wnt receptor encoding *frizzled* have

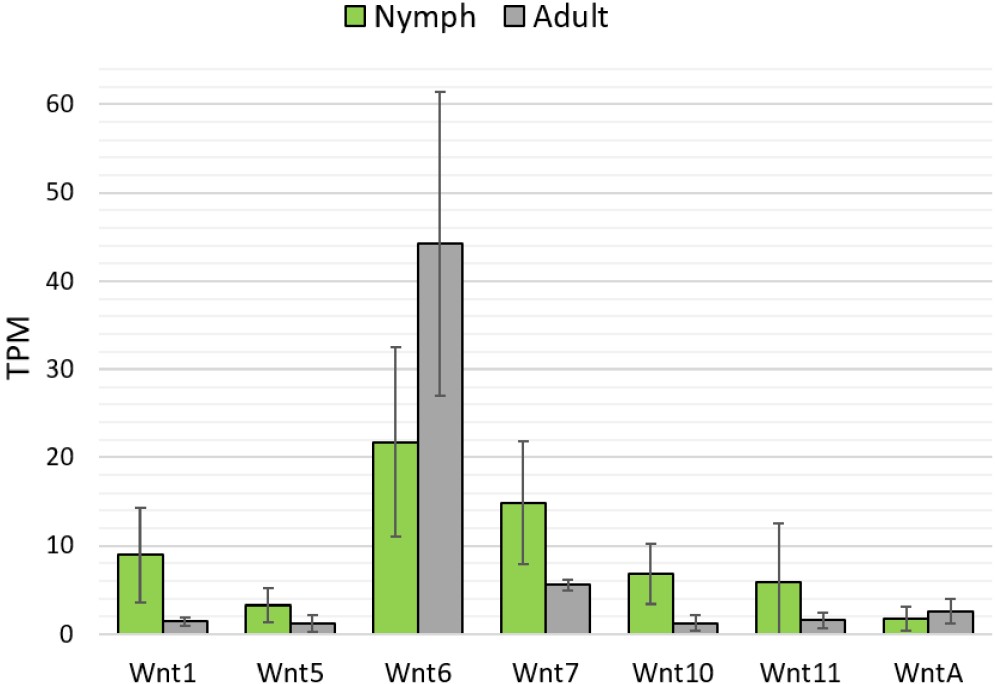

**Figure 7.** Transcript levels of *D. citri Wnt* repertoire in both nymph and adult psyllids from whole body RNA extractions. Green bars indicate the average transcript levels for *Wnt* in nymph samples [47], and gray bars represent the average transcript levels for *Wnt* in adult samples. Averages are based on six nymph samples and six adult samples. Expression levels shown in transcripts per million (TPM). Standard deviation of samples is shown by error bars. RNA-seq data was collected from CGEN [48].

been found in *D. citri*. We classified and numerically designated *D. citri's* three *frizzled* genes based on how their encoded protein sequences form clades with *D. melanogaster* orthologs (Figure 8). Our analysis showed that *D. citri*, and other hemipterans such as *Halymorpha halys* and *N. lugens*, possess a Frizzled protein like that of *D. melanogaster*'s Frizzled 3. Some hemipteran Frizzled orthologs form a distinct clade separate from the Dipteran sequences (Figure 8). The hemipteran clade suggests that these genes might belong to a different subfamily of Frizzled, maybe one specific to Hemiptera. However, this ortholog has not been reported in the *A. pisum* genome [22].

Orthologs for both *ROR1* and *ROR2* have been identified. Interestingly, *ROR1* has two isoforms, the first of which contains an immunoglobulin (IG) domain that is lacking from isoform 2 (Figure 9). *ROR1* isoform 2 (Dcitr05g14430.1.2) appears to average higher transcript levels in *D. citri* egg, nymph, and adult tissues than *ROR1* isoform 1 (Dcitr05g14430.1.1) based on CGEN data (Figure 10). Many transcripts for isoform 2 were detected in the psyllid egg (Figure 10). This suggests that expression of isoform 2 may be important in the early developmental stages of *D. citri*.

## Conclusion

Controlling the spread of *D. citri* is an important strategy for reducing the spread of HLB. With this study, we hope to provide a greater insight into *D. citri* biology, as well as accurate gene models that can be used in future research and applications. We have curated a comprehensive repertoire of *Wnt* signaling genes in *D. citri*. In total, 24 gene models

**Table 2.** Evidence supporting gene annotation. Manually annotated Wnt pathway genes in *Diaphorina citri*. There are 24 gene models in total. Each gene model has been assigned an identifier, and the evidence used to validate or modify the structure of the gene model has been listed. MCOT transcriptome identifiers that best support the manual annotation are also listed. The table is marked with an 'X' when supporting evidence of *de novo* transcriptome, Iso-Seq, RNA-Seq and ortholog support is present. MCOT: comprehensive transcriptome based on genome MAKER, Cufflinks, Oases, and Trinity transcript predictions; MAKER: gene predictions; *De novo* transcriptome: an independent transcriptome using Iso-Seq long-reads and RNA-Seq data; Iso-Seq transcripts: full-length transcripts generated with Pacific Biosciences technology; RNA-Seq: reads mapped to genome are also used as supporting evidence for splice junctions; Ortholog evidence: proteins from related hemipteran species and *Drosophila melanogaster*.

| Gene | OGS Identifier | MCOT | *de novo* transcriptome | Iso-Seq | RNA-Seq | Ortholog |
|---|---|---|---|---|---|---|
| *Wnt1* | Dcitr04g11660.1.1 | MCOT05703.0.CO | X | | X | X |
| *Wnt5* | Dcitr13g03650.1.1 | MCOT16538.0.CO | X | | X | |
| *Wnt6* | Dcitr04g11650.1.1 | MCOT21516.1.CO | X | X | X | |
| *Wnt7* † | Dcitr13g03730.1.1 | MCOT12704.0.CO | X | | X | X |
| *Wnt10* | Dcitr04g11640.1.1 | MCOT09136.0.MO | X | | X | X |
| *Wnt11* | Dcitr09g05250.1.1 | MCOT15024.0.CT | X | | X | |
| *WntA* | Dcitr13g02920.1.1 | MCOT02236.1.CT | X | | X | X |
| *Pangolin* † | Dcitr06g15680.1.1 | MCOT15454.2.CC | X | | X | |
| *Armadillo* | Dcitr10g09220.1.1 | MCOT18153.0.CT | | X | X | X |
| *Wntless* | Dcitr01g07340.1.1 | MCOT02320.0.CC | X | X | X | X |
| *Porcupine* | Dcitr13g04750.1.1 | MCOT19771.0.CO | X | X | X | |
| *Derailed* | Dcitr01g12220.1.1 | MCOT04433.1.CO | X | X | X | |
| *Doughnut* | Dcitr01g07650.1.1 | MCOT18207.0.CT | X | X | X | X |
| *Arrow* | Dcitr11g02670.1.1 | MCOT01906.1.CO | X | X | X | X |
| *Frizzled* | Dcitr04g04630.1.1 | MCOT11925.0.MO | X | | X | |
| *Frizzled 2* | Dcitr10g03570.1.1 | MCOT07682.0.MO | X | X | X | |
| *Frizzled 3* | Dcitr01g12100.1.1 | MCOT03353.0.CC | X | X | | |
| *ROR1* | Dcitr05g14430.1.1 | MCOT18375.0.CT | X | X | X | X |
| | Dcitr05g14430.1.2 | MCOT01992.1.CT | | | | |
| *ROR2* | Dcitr08g10450.1.1 | MCOT22482.0.CC | X | X | X | X |
| *Dishevelled* | Dcitr01g03830.1.1 | MCOT11762.0.MO | X | | X | X |
| *Shaggy* | Dcitr03g15060.1.1 | MCOT06728.0.CT | X | X | X | X |
| *Axin* | Dcitr07g09620.1.1 | MCOT05716.1.CT | X | | X | |
| *ck1-gamma* | Dcitr11g04200.1.1 | MCOT05782.2.CO | X | X | X | X |
| *Apc*-like | Dcitr07g12790.1.1 | MCOT14853.2.CO | X | | X | |

†Gene is manually annotated as a partial model in Genome v3.0. A complete representation of the gene and protein sequence can be determined with MCOT transcriptome data.

corresponding to canonical and noncanonical Wnt signaling have been annotated. The mechanisms of Wnt signaling appear to be mostly conserved and comparable to those found in *D. melanogaster* and other insects. These findings provide a greater insight into the evolutionary history of *D. citri* and Wnt signaling in this important hemipteran vector. Manual annotation and an improved genome assembly with chromosomal length scaffold were essential for identifying high quality gene models.

## REUSE POTENTIAL

The manually curated genes will be included in the Citrus Greening Expression Network (CGEN) [48] as a part of the Official Gene Set version 3. This visualization tool is useful for understanding psyllid biology and comparative analysis because it contains public transcriptomics data for *Diaphorina citri* from various tissues, life stages, CLas infection levels and citrus hosts. Future work could utilize these gene models in developing CRISPR and RNAi systems that target and disrupt critical biological processes in *D. citri*, thus controlling the spread of HLB. This work was done as part of a collaborative community annotation project [49].

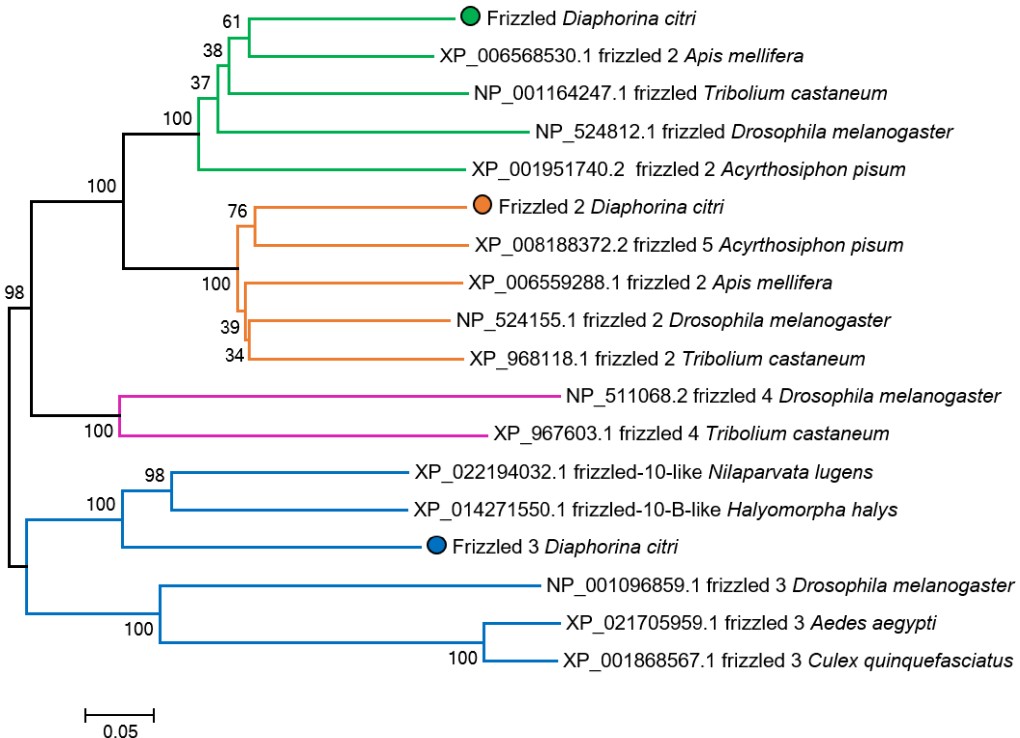

**Figure 8.** Neighbor-joining tree of insect Frizzled protein sequences. Proteins grouped in the Frizzled 1 subfamily are highlighted in green, Frizzled 2 in orange, Frizzled 3 in blue, and Frizzled 4 in magenta. Circles indicate the *D. citri* sequences. Ortholog sequences were collected from the NCBI protein database [18]. Some NCBI sequences (such as XP_006568530.1, XP_008188372.2, and XP_022194032.1) may have numeric labels derived from computational predictions that do not reflect sequence or functional similarity. Analysis was performed using MEGA7 [23].

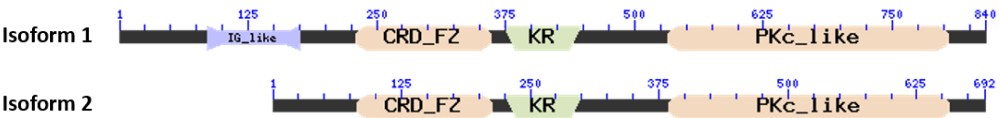

**Figure 9.** Domain comparison of *ROR1* isoforms. The immunoglobulin domain (IG_like) is present in isoform 1. Other shared domains include a cysteine-rich frizzled domain (CRD_FZ), a Kringle domain (KR), and a protein kinase catalytic domain (PKc_like). Domains were calculated and visualized using the NCBI Conserved Domain Architecture Retrieval Tool (CDART).

## DATA AVAILABILITY

Our annotation and gene curation workflow is described by Shippy *et al.* [17]. The *Diaphorina citri* genome assembly, official gene sets, and transcriptome data are accessible on the Citrus Greening website [50]. All accessions for genes used for phylogentic analysis are provided within this report (Tables 2, 3, Figure 8). We have included the Newick and Multiple Sequence Alignment files used to construct the Wnt neighbor-joining phylogenetic tree and other data is available in the *GigaScience* GigaDB repository [51].

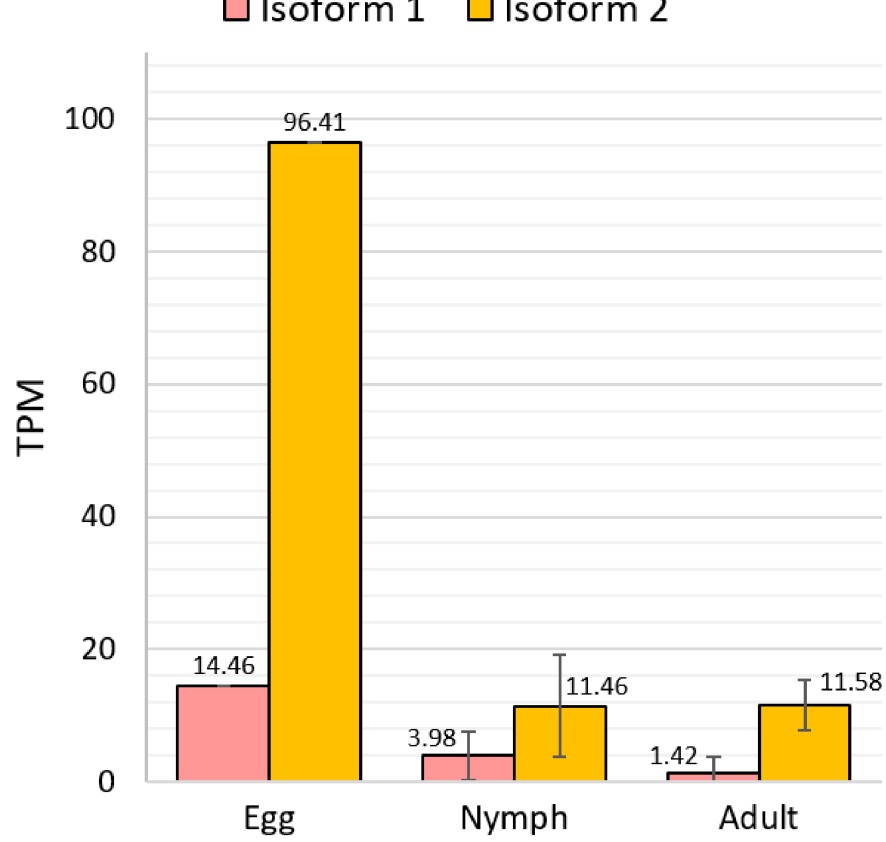

**Figure 10.** Expression of *ROR1* isoforms in egg, nymph and adult *D. citri*. Pink bars indicate the average transcript levels for isoform 1 (Dcitr05g14430.1.1), and orange bars indicate the average transcript levels for isoform 2 (Dcitr05g14430.1.2). Note: only one egg sample was used for comparison. Egg transcripts from the whole egg (one sample), nymph transcripts from the whole body (six samples), and adult transcripts from the whole body, abdomen, and thorax (14 samples) are shown. Expression values shown in transcripts per million (TPM). Data labels note the average TPM. Standard deviation of samples, when available, is shown by error bars. RNA-seq data was collected from CGEN [48].

## EDITOR'S NOTE

This article is one of a series of Data Releases crediting the outputs of a student-focused and community-driven manual annotation project, curating gene models and – if required – correcting assembly anomalies, for the *Diaphorina citri* genome project [46].

## DECLARATIONS
## LIST OF ABBREVIATIONS

CGEN: Citrus Greening Expression Network; HLB: Huanglongbing; MCOT: Maker, Cufflinks, Oases, and Trinity; NCBI: National Center for Biotechnology Information

## ETHICAL APPROVAL

Not applicable.

## CONSENT FOR PUBLICATION

Not applicable.

**Table 3.** Accessions for Wnt phylogenetic tree.

| NCBI accession | Species | NCBI protein name | Referred to in Figure 3 as |
|---|---|---|---|
| XP_002609873.1 | *Branchiostoma floridae* | Hypothetical protein BRAFLDRAFT_60204 | WntA |
| XP_024085687.1 | *Cimex lectularius* | Wnt-8b-like | Wnt8 |
| XP_014257242.2 | *Cimex lectularius* | Wnt-7b isoform X1 | Wnt7 |
| NP_476972.2 | *Drosophila melanogaster* | Wnt oncogene analog 4 isoform A | Wnt9 |
| NP_476924.1 | *Drosophila melanogaster* | Wnt oncogene analog 5 isoform A | Wnt5 |
| NP_476810.1 | *Drosophila melanogaster* | Wnt oncogene analog 2 isoform A | Wnt7 |
| NP_609109.3 | *Drosophila melanogaster* | Wnt oncogene analog 10 | Wnt10 |
| NP_609108.3 | *Drosophila melanogaster* | Wnt oncogene analog 6 isoform B | Wnt6 |
| NP_523502.1 | *Drosophila melanogaster* | Wingless | Wnt1 |
| NP_650272.1 | *Drosophila melanogaster* | wnt inhibitor of dorsal | Wnt8/D |
| ALO81632.1 | *Penaeus vannamei* | Wnt-16 | Wnt16 |
| OXA45577.1 | *Folsomia candida* | Wnt-16 | Wnt16 |
| XP_025422997.1 | *Sipha flava* | Wnt-16-like | Wnt16 |
| XP_022821085.1 | *Spodoptera litura* | Wnt-4-like | Wnt9 |
| XP_015835609.1 | *Tribolium castaneum* | Wnt-4 | Wnt9 |
| XP_008196351.1 | *Tribolium castaneum* | Wnt-7b isoform X1 | Wnt7 |
| XP_008195370.1 | *Tribolium castaneum* | Wnt-1 | WntA |
| XP_015835988.1 | *Tribolium castaneum* | Wnt-11b-1 isoform X1 | Wnt11 |
| XP_008193179.1 | *Tribolium castaneum* | Wnt-10a isoform X1 | Wnt10 |
| NP_001164137.1 | *Tribolium castaneum* | Wnt6 protein precursor | Wnt6 |
| NP_001107822.1 | *Tribolium castaneum* | wingless precursor | Wnt1 |
| XP_974684.1 | *Tribolium castaneum* | Wnt-5b | Wnt5 |
| XP_971439.1 | *Tribolium castaneum* | Wnt-8a isoform X1 | Wnt8 |
| XP_021702998.1 | *Aedes aegypti* | Wnt-4 | WntA |
| XP_557821.3 | *Anopheles gambiae* | AGAP008678-PA | WntA |
| XP_006561993.1 | *Apis mellifera* | Wnt-5b isoform X1 | Wnt5 |
| XP_006557287.1 | *Apis mellifera* | Wnt-7b isoform X1 | Wnt7 |
| XP_006567803.2 | *Apis mellifera* | Wnt-11b | Wnt11 |
| XP_016771882.1 | *Apis mellifera* | Wnt-6 isoform X1 | Wnt6 |
| XP_026300091.1 | *Apis mellifera* | Wnt-1 | Wnt1 |
| XP_396944.4 | *Apis mellifera* | Wnt-10b | Wnt10 |
| XP_001949667.2 | *Acyrthosiphon pisum* | Wnt-5b | Wnt5 |
| XP_016664156.1 | *Acyrthosiphon pisum* | Wnt-16 | Wnt16 |
| XP_001948541.2 | *Acyrthosiphon pisum* | Wnt-2 | Wnt7 |
| XP_001947400.1 | *Acyrthosiphon pisum* | Wnt-1 | WntA |
| XP_001944637.3 | *Acyrthosiphon pisum* | Wnt-11b-like isoform X1 | Wnt11 |
| XP_001945295.1 | *Acyrthosiphon pisum* | Wnt-1 | Wnt1 |
| XP_022184533.1 | *Nilaparvata lugens* | Wnt-16-like | Wnt16 |
| XP_022188550.1 | *Nilaparvata lugens* | Wnt-7b | Wnt7 |
| BAB62039.1 | *Homo sapiens* | WNT5B | Wnt5B |
| NP_003382.1 | *Homo sapiens* | Wnt-2 precursor | Wnt2 |
| NP_057171.2 | *Homo sapiens* | Wnt-16 isoform 2 | Wnt16 |
| NP_004616.2 | *Homo sapiens* | Wnt-7a precursor | Wnt7a |
| NP_478679.1 | *Homo sapiens* | Wnt-7b precursor | Wnt7b |
| NP_004617.2 | *Homo sapiens* | Wnt-11 precursor | Wnt11 |
| NP_003386.1 | *Homo sapiens* | Wnt-9a precursor | Wnt9a |
| NP_003387.1 | *Homo sapiens* | Wnt-9b isoform 1 precursor | Wnt9b |
| NP_110388.2 | *Homo sapiens* | Wnt-4 precursor | Wnt4 |
| NP_079492.2 | *Homo sapiens* | Wnt-10a precursor | Wnt10a |
| NP_003385.2 | *Homo sapiens* | Wnt-10b precursor | Wnt10b |
| NP_006513.1 | *Homo sapiens* | Wnt-6 precursor | Wnt6 |
| NP_005421.1 | *Homo sapiens* | proto-oncogene Wnt-1 precursor | Wnt1 |
| NP_001287867.1 | *Homo sapiens* | Wnt-8a isoform 1 precursor | Wnt8 |

## COMPETING INTERESTS

The authors declare that they have no competing interests.

## AUTHORS' CONTRIBUTIONS

WBH, SJB, TD and LAM conceptualized the study; TD, SS, TDS and SJB supervised the study; SJB, TD, SS, and LAM contributed to project administration; CV, MR, and RN conducted investigation; PH, MF-G, and SS contributed to software development; SS, TDS, PH, and MF-G developed methodology; SJB, TD, WBH, and LAM acquired funding; CV prepared and wrote the original draft; TD, SS, TDS, and SJB reviewed and edited the draft.

## FUNDING

This work was supported by USDA-NIFA grants 2015-70016-23028, HSI 1300394 and 2020-70029-33199.

## ACKNOWLEDGEMENTS

We thank Alistair McGregor and Joshua Benoit for valuable discussions.

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
