## [Reviewer Report]

Comments on revised manuscriptYes. My queries have been addressed, and most importantly, the data I requested has been uploaded.

---

## [Reviewer Report]

Reviewer name and names of any other individual's who aided in reviewer Mary Ann TuliDo you understand and agree to our policy of having open and named reviews, and having your review included with the published papers. (If no, please inform the editor that you cannot review this manuscript.)YesIs the language of sufficient quality?YesPlease add additional comments on language quality to clarify if needed
The manuscript reads very well.Are all data available and do they match the descriptions in the paper? NoAdditional Comments1) Line 176. "High scoring MCOT models were then searched on the NCBI protein database...."
We need the list Wnt pathway genes with high scoring MCOT models.

2) Line 178. "The high scoring MCOT models that had promising NCBI search results were used to search the D. citri assembled genome."
We need the list of high scoring MCOT models which had promising NCBI search results..

3) Line 179. "Genome regions of high sequence identity to the query sequence were investigated within JBrowse"
We need the list of models with high sequence identity with the assembled genome.

4) Line 184. "MUSCLE multiple sequence alignments of the D. citri gene model sequences and orthologous sequences were created through MEGA7"
We need the output of MUSCLE (FASTA). 
We need the files underlying the phylogenetic tree (newick).

5) I note that MEGA7 has been used. I wonder why the newer release (MEGAX, March '21) was not used. Furthermore, the annotation protocol (dx.doi.org/10.17504/protocols.io.bniimcce) suggests using Mega7 or MegaX.

Instructions on how to upload these files is given under "Any Additional Overall Comments to the Author".
Are the data and metadata consistent with relevant minimum information or reporting standards? See GigaDB checklists for examples <a href="http://gigadb.org/site/guide" target="_blank">http://gigadb.org/site/guide</a>YesAdditional CommentsNomenclature standards have been met.
All cited INSDC accession numbers are publicly available.
Is the data acquisition clear, complete and methodologically sound?YesAdditional CommentsCuration workflow used for community annotation is available via protocols.io , nonetheless the manuscript includes comprehensive summary which is appropriate.Is there sufficient detail in the methods and data-processing steps to allow reproduction?NoAdditional CommentsSee "Are all data available and do they match the descriptions in the paper?" above.
Once the additional files are made available I believe reproduction will be possible.Is there sufficient data validation and statistical analyses of data quality? YesAdditional CommentsIs the validation suitable for this type of data?YesAdditional CommentsIs there sufficient information for others to reuse this dataset or integrate it with other data?YesAdditional CommentsAny Additional Overall Comments to the AuthorSome of my comments/recommendations are pertinent to the other D. citri manuscripts currently under review.

Please send the files to us via the FTP server details at the end of this email.
Please note the following.

We strongly recommend filenames are concise and meaningful (e.g. "gene_exp_cell_lines.csv" is better than "Supp1.txt").
Filenames should be unique.
Filenames should not include spaces. We recommend using the underscore (_) in place of spaces in the filenames.
Filenames should only include the following characters a-z,A-Z,0-9,_,-,+,.
Text and tabular information must be in a non-proprietary and text-based format (e.g. CSV/TSV rather than PDF or XLS)
Images must be in a lossless raster format (e.g. TIFF, PNG) or vector format (e.g. SVG).
If you encounter any errors please send us the full context and error message to help us resolve the problem.

username = user96
password = Sahacitri
FTP server = parrot.genomics.cn

For using tools like FileZilla use the standard FTP protocol (not sftp).

If you are using command line FTP, you may need to use the passive mode (e.g. use epsv command):

> ftp <username>@parrot.genomics.cn
Connected to parrot.genomics.cn.
220 (vsFTPd 2.0.5)
331 Please specify the password.
Password:
230 Login successful.
Remote system type is UNIX.
Using binary mode to transfer files.
ftp> epsv
EPSV/EPRT on IPv4 off.
ftp> put <file_name>
local: <file_name> remote: <file_name>
227 Entering Passive Mode (218,188,108,76,126,77)
150 Ok to send data.
100% |************************************************************************| 12 137.86 KiB/s 00:00 ETA
226 File receive OK.
12 bytes sent in 00:00 (0.26 KiB/s)
ftp>RecommendationMinor Revision

---

## [Reviewer Report]

Reviewer name and names of any other individual's who aided in reviewer Xingtan Zhang, Dongna MaDo you understand and agree to our policy of having open and named reviews, and having your review included with the published papers. (If no, please inform the editor that you cannot review this manuscript.)YesIs the language of sufficient quality?YesPlease add additional comments on language quality to clarify if needed
Are all data available and do they match the descriptions in the paper? YesAdditional CommentsAre the data and metadata consistent with relevant minimum information or reporting standards? See GigaDB checklists for examples <a href="http://gigadb.org/site/guide" target="_blank">http://gigadb.org/site/guide</a>YesAdditional CommentsIs the data acquisition clear, complete and methodologically sound?YesAdditional CommentsIs there sufficient detail in the methods and data-processing steps to allow reproduction?YesAdditional CommentsIs there sufficient data validation and statistical analyses of data quality? YesAdditional CommentsIs the validation suitable for this type of data?YesAdditional CommentsIs there sufficient information for others to reuse this dataset or integrate it with other data?YesAdditional CommentsAny Additional Overall Comments to the AuthorThe manuscript by Vosburg et. al., systematically analyzed of the characteristics of the Wnt signaling genes in Diaphorina citri, and focusing on evolutionary history, expression patterns and potential functional. Finally, they also performed manual annotation of the Wnt signaling pathway. Indeed, this work would add important resource for the study of the evolutionary history of D. citri and Wnt signaling in this important hemipteran vector. The writing is acceptable. Even though, I still have some suggestion which may improve this manuscript.

1. In the methods, the authors have indicated the process of identifying win genes, but the abstract describes it as Curation identification? I am confused whether this Wnt signaling genes in D. citri were identified by the author or whether the author just further analyzed it using the results already identified by others?
2. The paper just did the identification of the win gene, evolutionary, and then the expression analysis using RNA-seq. It is recommended to also look at the chromosomal localization and mode of origin (e.g., tandem repeats)
3. The Wnt signaling genes related to the hemipteran vector studied by the authors can be further verified by qPCR and then compared with the expression and function of other published insect-related genes for discussion.
RecommendationMajor Revision